# The Impact of Climate Change and Human Activity on Spatiotemporal Patterns of Multiple Cropping Index in South West China

**Chuangjuan Zhang [1],[†]** , **Hongming He [1,2,3],*** **and Ali Mokhtar [1,2,4],†**

[1]   State Key Laboratory of Soil Erosion and Dryland Farming on Loess Plateau, Institute of Soil and Water Conservation, Northwest Agriculture and Forestry University, Yangling 712100, China; chuangjuan@nwafu.edu.cn (C.Z.); Ali.mokhtar@agr.cu.edu.eg (A.M.)

[2]   Institute of Soil and Water Conservation, Chinese Academy of Sciences and Ministry of Water Resources, Yangling 712100, China

[3]   School of Geographic Sciences, East China Normal University, Shanghai 210062, China

[4]   Department of Agricultural Engineering, Faculty of Agriculture, Cairo University, Giza 12613, Egypt

*   Correspondence: hongming.mobile@gmail.com; Tel.: +86-29-87012884 (ext. 712100)

†   These two authors contributed equally.

**Abstract:** Agricultural lands are very sensitive to climate and human activity changes, which result in variations in regional agricultural resources and decreased production of total grain output and increased difficulty in producing grain yields. Multiple cropping is one of the simplest ways to increase grain production. The research aims is to analyze the spatial and temporal variations in the multiple cropping index and study the factors that influence the multiple cropping index. Based on the maximum multiple cropping index (MCI) and a "heat-precipitation" quantitative relation model, we analyzed the theoretical potential multiple cropping index (PMCI) and the spatiotemporal changes in the potential increase in the multiple cropping index (PIMCI). Our results are as follows: The MCI was significantly higher in the eastern region than in the western region and higher in the central region than in the northern and southern regions; in Yunnan Province, it showed a fluctuating downwards trend; further, it exhibited sudden declines from 2004 to 2006 and from 2012 to 2014 in Guizhou, while it exhibited an increasing trend in Sichuan Province. The PMCIs were the highest in the eastern and southern regions, especially in eastern Sichuan Province, and the PIMCI was significantly higher in Yunnan Province than in Guizhou and Sichuan. Climate change, human activities, and terrain had significant influences on the MCI changes in southwest China, especially the temperature change, which was the key factor affecting the MCI changes. The dominant land use types in southwest China were forest (46%), grass (28%), and farmland (23%) during 1980–2015. Therefore, the adjustment of the planting structure in different terrain areas according to the temperature changes has become the main strategy to promote the sustainable development of cultivated land resources in the region, further, the results would help implement the plan to increase grain production capacity in southwest China.

**Keywords:** climate change; population growth; grain production; degree of land intensive use; spatiotemporal pattern; land use cover change

---

## 1. Introduction

China supplies food to 22% of the world population with only 7% of the world's cropland base. Therefore, China's food demands seriously conflict with the supply [1]. Agricultural lands are very sensitive to changes in climate and human activity, which result in variations in regional agricultural

resources and production [2,3]. With the rapid development of industrialization and urbanization, large amounts of cultivated land resources have been used for non-agricultural purposes. Humans need farmland to survive and it plays a major role in food production and food security [4,5]. Land is not the only factor that affects agricultural production [6]. Grain production increases would come more from intensified use of existing land rather than from increasing the amount of land under cultivation in the future [7,8]. In the past, by improving the production of crops and optimizing the management of production facilities and advanced technologies, several studies have assessed realistic yield potentials with the goal of ensuring increases in grain output from a stable cultivated area [9,10]. However, as the crop yield per unit area continuously improves, the difficulty in increasing the yield per unit area gradually increases as the maximum yield is approached; as a result, output growth has stalled in many places [11,12]. Cropping intensity is often measured using a multiple cropping index; multiple cropping index (MCI), the number of crops planted per year in the same piece of land, representing the actual situation of the multiple cropping systems, multiple cropping is the most direct and effective way to increase regional grain production, further, it is one of the simplest ways to increase grain production [13,14]. Therefore, this method can mitigate the human pressure on farmland resources by limiting the cultivated land resources that are needed via increasing crop production [15,16]. A previous study found that 37.55% of the cultivated land area throughout the world could implement multiple cropping [17]. References [8,18] considered the multiple cropping index as one of the important crop land characteristics for evaluating the food security of China. They concluded that making the regional difference of MCI clear could support policy making. For example, it was found that the multiple cropping index (MCI) of cultivated land increased by 9.7% in China, which was equivalent to an increase of 85,000 km$^2$ of crop planting area, which could increase the annual grain yield by 24.1 billion ha, accounting for more than 36.5% of the increased grain yield. Furthermore, the MCI has become the main method for increasing grain production, accounting for more than 30% of the total grain output of China [14,19]. Cropping intensity may have significant impact on irrigation water use, biogeochemical cycles and climate, moreover, it varies substantially year by year weather and climate, regional and global markets of crop production, and individual famers' decisions on crop cultivation [20].

The MCI is the observed multiple cropping value that occurs in real cultivated land, and it is usually used to measure the degree of intensive utilization of cultivated land resources and evaluate the basic situation of the utilization of cultivated land resources. The research on the complex species index can be roughly divided into two types: one is the calculation of the cultivated land replanting index based on statistical data, and the other is the monitoring of the plant net primary productivity based on remote sensing SPOT-NDVI data [12,21–23]. In terms of research content, the existing studies have mainly focused on the evaluation of the contribution of the MCI to grain yield and the characteristics of the temporal variations in the MCIs. For example, in China, it was estimated that the MCI increased from 131% to 158% from 1952–1995 [11], which represented an increase of 27% during this period, and the multi-cropping croplands produced approximately 75% of the country's grain [24]. Food security is significantly impacted by the contribution of the MCI to the Chinese grain yield potential. Subsequently, some researchers have used statistical data to analyze the variations in the MCI over long periods of time and concluded that the MCI of cultivated land increased gradually from North to South China; additionally, the MCIs of the 31 provinces in China differed significantly in terms of space [25]. The potential multiple cropping index (PMCI) is the maximum MCI that can be achieved by making full use of natural resources such as water, soil, light, and heat in multiple cropping patterns. PMCI is the maximum number of crops planted per year in the same piece of land, representing the potential capacity for the implementation of multiple cropping. [14]. Recent research has actively engaged in spatially exploring MCI and potential MCI (PMCI) at various geographic scales using different kinds of data, in order to assess the current situation of multiple cropping systems in China [14]. Multiple cropping not only improves the utilization rate of natural and human resources but also alleviates the conflict between food crops and cash crops by adjusting the planting structure of crops [26,27]. A previous study estimated the maximum PMCI at a global 10 km spatial scale

grid using the temperature threshold method [12]. Subsequently, some researchers established an empirical estimation model of the "heat-precipitation" potential according to the correlation between the regional heat and water resources with the maximum MCI [28,29]. They also estimated the maximum PMCI in China by using the agro-ecological zone model and the stochastic frontier method, taking into account variables such as temperature, precipitation, soil, topography, and socioeconomic factors. From the perspectives of the study area and scale, existing studies have mainly focused on the main grain-producing areas, such as the North China Plain, central China, and the middle and lower reaches of the Yangtze River [30]. Reference [27] presented that the PMCI gap between rain-fed and irrigated scenarios increased from 18% to 24%, which indicated noticeable growth of water supply limitations under the rain-fed scenario. The MCE of China in 2005 was 87.5%, with a gap of 22% between the MCI and the PMCI [14]. In contrast, the current situation and problem of the MCI in southwest China indicates that the MCI of dry land is generally low, the development between regions is unbalanced and there are several unused fields in winter. Furthermore, in the 1990s, China entered a stage of comprehensive urbanization, and a large amount of cultivated land was occupied by construction land [31]. Although the state had introduced policies such as "returning farmland to forests and grasslands" and farmland protection, the amount of farmland was still decreasing quickly. Moreover, the quality and productivity of cultivated land continued to decline. In addition, agricultural production has emphasized the development of efficient economic crops and to some extent, it has tended to ignore grain production and compressed the planting area of double-cropping rice [32,33], which has resulted in the decline in the MCI in southwest China in recent years. Therefore, studies on the spatial and temporal changes in the MCI in this region have become an important part of the research on land use and cover changes in this region. In the past, the spatial and temporal variations in the MCI in southwest China were not comprehensively evaluated, so our research aim is to (1) estimate the PMCI of districts and evaluate the regional distribution of the potential increase in the multiple cropping index (PIMCI) in three provinces of southwestern China, (2) point out the factors influencing the MCI, and (3) present some agricultural policy strategies to develop the production conditions in order to increase the grain production and MCI in southwest China. Our findings can be used to properly regulate crop planting patterns under climate and human activity changes, optimize the allocation of resources, and provide a reference for developing agricultural policy.

## 2. Materials and Methods

### 2.1. Study Area

Southwest China is a typical grain-oil producing area with serious ecological degradations. The main crops are rice, winter wheat, spring corn, rape, and sugar cane. The output of rice (grain crop) and rapeseed (cash crop) accounts for 15.8% and 24.2% of the total national output, respectively. The climate in southwest China is quite different, ranging from tropical and subtropical to frigid zones, which form a vertical three-dimensional landscape with superior hydrothermal conditions and annual precipitation above 1000 mm [34]. In this area, the precipitation is evenly distributed in time and space, and the rain and heat are the highest in the same season; thus, crops can be harvested two or three times each year. However, due to the influence of the monsoon climate, topography (the elevation decreases from northwest to southeast) and other comprehensive factors, crops are mainly planted in mountainous areas and high-altitude areas, and these crops have weak resistance to natural disasters and unstable yields (Figure 1). For example, 70% of corn is planted in mountainous and high-altitude areas, where the production conditions are poor and natural disasters are frequent [35]; additionally, the yield is not stable and exhibits large differences between years. In this area, the distributions of one crop per year, two crops per year and three crops per year represent one of the few multi-cropping areas in China. The region has abundant agricultural resources and favorable conditions that can increase the potential for multiple cropping and improve the advantages of cultivated land utilization. A considerable amount of cultivated land is located in rocky desertification areas, where the land is

broken and the cultivated layer is shallow, and these conditions make cultivation difficult. In addition, a certain amount of cultivated land is damaged by geological floods, making it difficult to restore cultivation. Therefore, making full use of the superior water and heat conditions and exploring the potential of replanting cultivated land are important approaches that should be considered when trying to alleviate the contradiction between population increase and land and to ensure regional food security in the case of limited arable land resources.

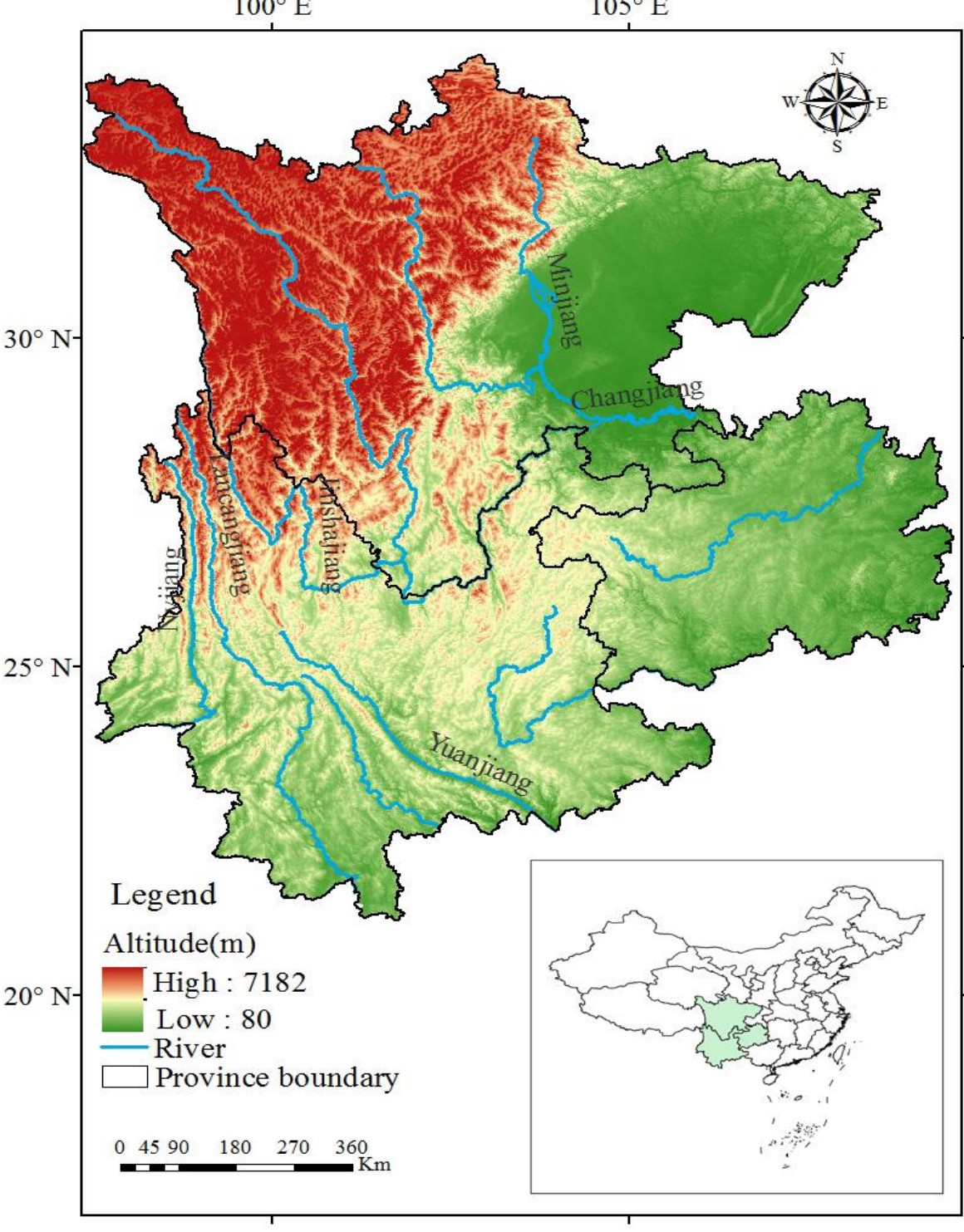

**Figure 1.** Location of the study area in southwestern China.

### 2.2. Data Set

The cultivated land data were mainly derived from the statistical yearbooks of three provinces in southwest China (Sichuan, Guizhou, and Yunnan Province). The land survey and revision data were from the Ministry of Land and Resources (http://www.yearbookchina.com/). The cultivated land data after 2012 were derived from the second national land survey. The sown area of crops, sown area of grain crops, yield of grain crops and other grain production data were derived from statistics on Chinese agriculture, which were published by the Ministry of Agriculture. The meteorological data were obtained from the daily surface climate data set of the China meteorological science data sharing service network (http://cdc.cma.gov.cn). The digital elevation model (DEM) data were obtained from the geospatial data cloud of the Computer Network Information Center, Chinese Academy of Sciences (http://www.gscloud.cn). The vector data of soil type (1:100,000) were obtained from the Data Center of Resource and Environmental Sciences, Chinese Academy of Sciences (http://www.resdc.cn). The agricultural survey data were the effective yearly data after screening the abnormal data in the cultivated land area and sown area data after 1978, and then the MCI was calculated, with a theoretical value between 0% and 300%. Daily data from the meteorological stations of three provinces in southwest China were transformed into annual data to calculate the annual precipitation and temperatures.

### 2.3. Statistical Analysis of Spatiotemporal Distribution Patterns

*The* Mann–Kendall (MK) test was carried out to analyze the annual and seasonal trends in the climate data. The purpose of the MK test is to statistically assess whether there is a monotonic upward or downtrend in the variable of interest over time [36]. The MK test does not require normality of the time series and is less sensitive to outliers and missing values. This test was recommended by the World Meteorological Organization (WMO), which applied the MK test to assess the significance of the changes in precipitation and temperature; this test has been widely used for the detection of hydrological trends in hydrological and meteorological data [37,38]. Pearson correlation coefficient analysis was used to explore the coupling analysis between two variables, to estimate the correlation between MCI and influence factors. We performed bilinear kriging interpolation for the meteorological data using ArcGIS and ArcView. The gridded chart of accumulated temperature and precipitation in the three provinces in southwest China was obtained using an image with a spatial resolution of 1 km.

### 2.4. Methods

The actual MCI is an approach used to increase the planting frequency of crops based on the original crop planting; it makes full use of idle fields and improves land use efficiency. The MCI is calculated on the administrative division unit using the statistics of crop planting area and cultivated land area [9], which is an observation of the real cropping situation. The calculation method is as follows:

$$MCI = \frac{A_S}{A_C} \times 100\% \tag{1}$$

where *MCI* is the actual multiple cropping index (%), $A_S$ is the total planting area of crops in a year (hm$^2$), and $A_C$ is the total area under cultivation (hm$^2$).

Water and heat are necessary for the complete growth and development of any crop; therefore, water and heat are the basic factors affecting the level of the MCI. In reality, the maximum MCI results from making full use of the available light, heat, and water resources. However, as the area with the fewest sunshine hours in China, some areas of the Sichuan Basin have implemented a tri-cropping system. Practice has proven that sunshine is not a meteorological factor that limits the improvement of MCIs. The PMCI was calculated with reference to the model of the quantitative relationship between the maximum MCI and heat and water resources [34], regardless of sunshine hours. According to Gao [39], the PMCI of cumulative temperature and the PMCI of precipitation were calculated using Equations (2) and (3), respectively. Finally, the minimum values of both the PMCI of precipitation and

the PMCI of cumulative temperature were collected for each pixel, and the PMCI was calculated using Equation (4).

$$M_T = \begin{cases} 100 & T < 3400 \\ (T-3400)\times 0.125+100 & 3400 \le T < 4200 \\ 200 & 4200 \le T < 5200 \\ (T-5200)\times 0.1+200 & 5200 \le T < 6200 \\ 300 & T > 6200 \end{cases} \tag{2}$$

$$M_R = \begin{cases} 100 & R < 500 \\ (R-500)\times 0.14+200 & 500 \le R < 1200 \\ 300 & R > 1200 \end{cases} \tag{3}$$

$$\text{PMCI} = MIN(MT, MR) \tag{4}$$

PMCI is the potential multiple cropping index, $M_T$ is the PMCI of cumulative temperature (%), and $M_R$ is the PMCI of precipitation (%). T is the accumulated temperature greater than 0 (°C). R is the average annual precipitation (mm).

$$\text{PIMCI} = \text{PMCI} - \text{MCI} \tag{5}$$

PIMCI is the difference between the PMCI and MCI, which was used to quantify the potential for multiple enhancements.

## 3. Results

### 3.1. The Change in Time Series of MCIs for Different Crops

The average annual MCI in southwest China exhibited a nonsignificant decreasing trend over the three provinces (Yunnan, Guizhou, and Sichuan) in different types of crops in the southwestern provinces of China, which is shown in Figure 2. The average rate of change in the MCI was 6.8%/10 year. Particularly, the MCI in Yunnan Province showed a fluctuating downward trend from 1990 to 2015 with a slope of −2.04%/year, with an average MCI of 145.38%, a maximum MCI of 178% in 1998, and a minimum MCI of 100.67% in 2010. In the period from 1990 to 2015, the MCI in Guizhou Province showed a slow increasing trend, reaching a maximum in 2010, with a value of 277.55% and a slope of 0.8%/year; however, the MCI exhibited sudden decreases from 2004–2006 and 2012–2014. From 1990 to 2015, the MCI in Sichuan Province increased significantly, reaching a maximum value of 241.94% in 2010, and the average MCI value in the whole province was 199.47%. After 2011, the minimum MCI value of 143.57% appeared in 2012, with the slope of the MCI being −1.3%/year due to the decrease in cultivated land. As seen in Figure 2b–d, the grain crop MCI was close to the change in the MCI. These changes were mainly divided into two stages: From 1990 to 1998, the grain crop MCI of Yunnan increased slowly (Figure 2b), and in 2000, the grain crop MCI of Yunnan decreased by 28% compared with that in the previous year, but it increased slowly after 2000 and increased by 6.58% in 2006. From 1990 to 2015, the grain crop MCI in Guizhou exhibited an increasing trend reaching a maximum of 175.29% in 2005 and then started to decrease in 2006 (Figure 2c). However, in the same period in Sichuan, the grain crop MCI grew slowly, with an average annual growth rate of 0.65% (Figure 2d). The greatest change in the variability of the grain crop MCI in the three provinces was from 2006 to 2015. The inter-annual variations of the grain crop MCI were as follows: Sichuan > Guizhou > Yunnan, with values of −41.01%, −33.55%, and −29.60%, respectively. The cash crop MCI has maintained a steady growth trend in recent years. The cash crop MCI of Yunnan Province decreased by 5.63% in 1994–1995; however, the overall cash crop MCI showed a double-peak stable growth pattern, and the maximum value of 48.39% appeared in 2012 (Figure 2b). The cash crop MCI of Guizhou Province showed fluctuating growth, with an average annual growth rate of 3.00%. Furthermore, the cash crop

MCI of Sichuan Province showed a single-peak growth pattern with an average annual growth rate of 1.88%.

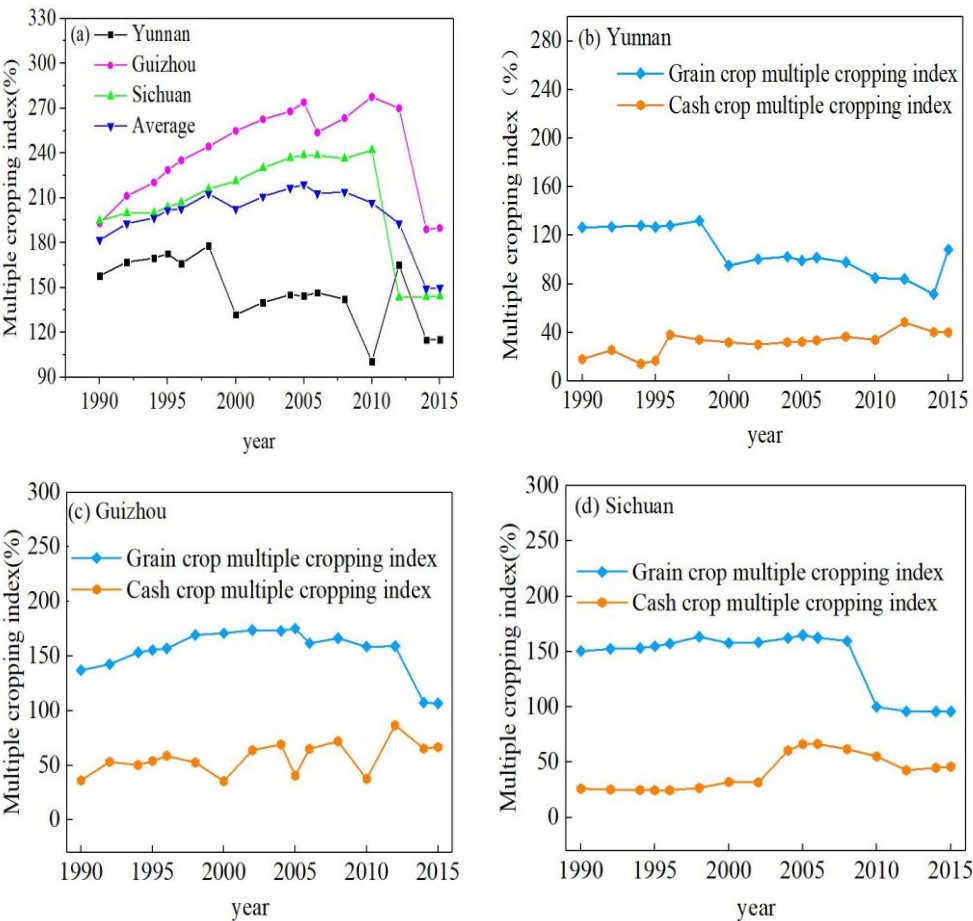

**Figure 2.** Interannual variation of the MCI and the MCI for different crop types in three provinces of southwest China.

*3.2. Spatial Distribution Characteristics*

3.2.1. Analysis of the Spatial Pattern Change in MCI

The spatial distributions of the MCI in the municipal administrative regions of southwest China during the past decades are shown in Figure 3. The MCI in the eastern region was significantly higher than that in the western region; moreover, the MCI in the central region was higher than that in the northern and southern regions. From the early 1990s to 2010–2015, the MCI in Guizhou Province showed the most significant growth, especially in Guiyang, Bijie, Qiannan, and Qiandongnan, with increases of 41.2%, 51.7%, 51.5%, and 80.9%, respectively, over the past 25 years, and the increases in the other municipal-level administrative regions ranged from 12.9% to 40.6%. From the early 1990s to the early 21st century, the MCI increased significantly in Guizhou Province, with the largest MCI values in Zunyi and Tongren county at 11.89% and 12.1%, respectively. The most obvious increase was in Qiannan county, where the MCI changed from 213.2% to 231.6%, increasing by approximately 13.84%. By 2010–2015, the MCI in Tongren county had declined by 17.9% compared with that in the early 21st century, while the MCIs of Bijie, Guiyang, Qiandongnan, and Qiannan were largest in 2010–2015 with values of 278.7%, 271.1%, 299.7%, and 283.7%, respectively. Compared with Guizhou, the largest MCI in Sichuan Province was mainly distributed in the eastern part of Sichuan in Bazhong, Dazhou, Nanchong, and Guang'an county, while the MCI was relatively low in western and northern Ganzi and Aba states, and there were no significant changes over the years due to the high altitude, severe climatic

conditions and long crop growth cycle. From the perspective of time series, the MCIs in Sichuan Province from 1990–1995, 2000–2005, and 2010–2015 were 188.34%, 212.32%, and 221.39%, respectively. Figure 3 shows that the change in the MCI in Guizhou was similar to that in Sichuan Province, which increased significantly in each region after the beginning of the 21st century. Compared with the early 1990s, in addition to the Ganzi Tibetan Autonomous Prefecture in the early 21st century, the MCI of all the other cities increased. In the past 10 years, the MCI increased by 1.04%~21.39%, with an average increase of 7.99%. Among them, Nanchong, Bazhong, and Guangyuan exhibited the greatest increases, with increases of 21.39%, 16.01%, and 15.42%, respectively. From the early 21st century to 2010–2015, the MCI declined in most parts of Sichuan Province, which were mainly distributed in Chengdu and its surrounding areas. The declines in Ya'an, Panzhihua, and Dazhou were the most obvious, reaching 64.34%, 15.32%, and 11.09%, respectively.

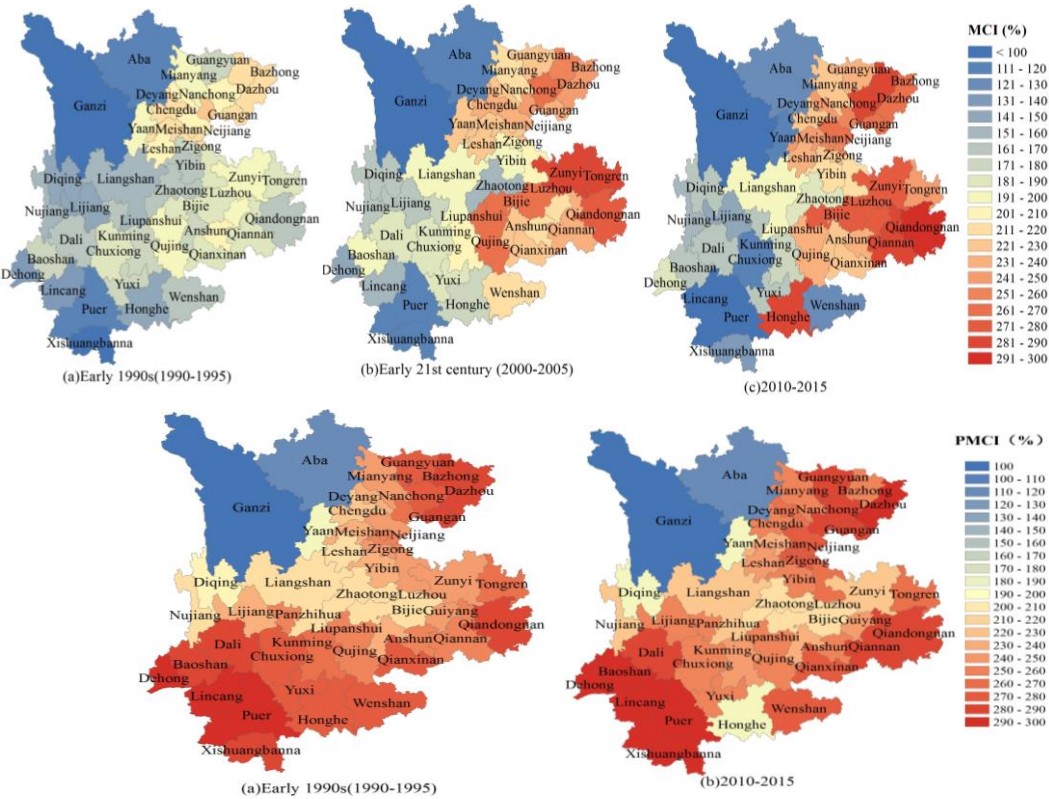

**Figure 3.** Distribution characteristics of the MCI and PMCI in southwest China.

In contrast, the MCI of Yunnan Province in the southern part of the southwest mountainous area exhibited relatively small changes over the past 25 years (Figure 3); however, the level of cultivated land replanting was lower than that in Guizhou and Sichuan. The MCIs of Yunnan Province from 1990–1995, 2000–2005, and 2010–2015 were 155.89%, 174.67%, and 163.72%, respectively. At the same time, the MCI of Yunnan Province reached the maximum value in the early 21st century, although the MCI of Yunnan decreased in 2010–2015, especially in Chuxiong and Lincang, with decreases of 37.15% and 35.65%, respectively. The most significant change in the MCI occurred in the city of Honghe in southeastern Yunnan, and the MCI increased 1.94 times from 147.94% in the early 1990s to 287.25% in 2010–2015, and this increase was much higher than that in the other cities. In general, from the early 1990s to 2010–2015, the MCI of various cities in Yunnan Province exhibited an increasing trend, especially in central Yunnan; however, the degree of intensive utilization was relatively low. The MCI in the southern region, represented by Xishuangbanna and Puer, was the lowest, and the phenomenon of abandoned farmland in the southern part of Yunnan was serious.

3.2.2. Analysis of the Spatial Pattern of the Change in the PMCI

Cultivated land and climatic resources are the basic conditions for determining the replanting pattern of a regional crop. Southwest China is dominated by hilly areas, and it has different climate conditions from tropical and subtropical to frigid zones, which form a vertical three-dimensional landscape. Therefore, climate resources are the main constraint for the PMCI in this region. This research was based on the data of temperature and precipitation changes in the southwestern region in the early 1990s and in recent years. Using the model of the quantitative relationship between the maximum MCI and heat and water resources, this paper compared and analyzed the theoretical PMCI of the two periods, i.e., the early 1990s (1990–1995) and 2010–2015. Based on the spatial patterns of the PMCI in the three provinces of southwestern China, in the early 1990s and 2010–2015, the PMCI of southwest China was significantly higher in the eastern and southern regions than in the central and northwestern regions (Figure 3). Overall, the spatial differences in the PMCI in southwest China were obvious, and the time differences were not significant. Particularly, in the early 1990s, the lowest PMCI values in Sichuan Province were found in Ganzi and Aba County. Further, the largest values were found in the eastern region of Sichuan, such as in Bazhong, Dazhou, and Guang'an, with potential values of 288.6%, 290%, and 286.3%, respectively, and the values in the eastern region were much higher than those in the western region. The remaining central urban PMCI fluctuated between 200% and 277.5%. From the early 1990s to 2010–2015, as the climate continued to warm, there were no significant changes in the PMCI values of Ganzi, Aba, and Ya'an in the western part of Sichuan, while the average PMCI of Bazhong in the eastern part of Sichuan showed a decreasing trend, with the potential value reduced by 11.1%. The PMCI in the other areas of Sichuan showed an increasing trend, increasing by 1.1%~12.7%. In the early 1990s, the PMCI was relatively high in all regions of Yunnan; among these regions, the PMCIs of Dehong, Lincang, and Puer in the west of Yunnan Province were the highest, with values up to 300%. There were large areas where light, heat, and water resources could be fully utilized in this region. In contrast, the PMCI in Diqing city was the lowest, and it was the only region in the province where crops were grown once a year. The PMCI in Zhaotong was 211.3%, while the PMCI in the other regions ranged from 238.8% to 288.6%. In 2010–2015, except in Dehong, Lincang, and Puer, which maintained high PMCIs, the PMCIs of Baoshan and Xishuangbanna reached a maximum of 300%. Overall, the PMCI in the eastern part of Yunnan showed a declining trend, especially in Honghe, where the PMCI value was 30.8% lower than that in the early 1990s. Furthermore, during the early 1990s and 2010-2015, in Guizhou Province, except for the obvious changes in the PMCI of Qiannan and Zunyi, the changes in the PMCIs in other cities were not significant over the past 25 years. In both periods, the PMCIs in Qiandongnan were the largest, reaching 281.1% and 280.1%, respectively.

3.2.3. Analysis of the Spatial Pattern of the Change in the PIMCI

The change in the PIMCI in southwest China during the early 1990s and 2010–2015 indicated that the PIMCI value of Yunnan Province was significantly higher than that of Guizhou and Sichuan provinces. In the early 1990s, Aba and Yaan of Sichuan Province were regions with no potential increases, while Ganzi, Deyang, Chengdu, Meishan, and Leshan were regions with low potential increases, and the improvements ranged from 0 to 25%. Similar to the PMCI, the PIMCI was larger in the Guangyuan, Bazhong, Dazhou, and Guang'an regions in eastern Sichuan, among which the largest PIMCI was 70.4% in Guangyuan (Figure 4). As shown in Figure 4b,c, in 2010–2015, compared with the early 1990s, Aba changed from an area with no potential to an area with low potential; however, the potential for improvement was only 0.77%. Bazhong city suddenly changed from an area with high potential to an area with no potential, and the potential values of Dazhou, Guang'an, Neijiang, and Ziyang declined. First, in the early 1990s, except for Diqing and Qujing, which were the only areas with medium potential, the re-cultivation of land in the other areas of Yunnan Province produced areas with high potential. Among them, the largest PIMCI was in Pu'er city, at 173.5%. However, in 2010–2015, Qujing city declined to an area with low potential, and Honghe city changed from having high potential ten years ago to no potential. The rest of the cities did not change significantly;

similarly, the PIMCI in Pu'er city was the largest. In the early 1990s, Qiandongnan and Qianxinan of Guizhou Province were areas with high potential. Bijie city had no potential for improvement, and the remaining cities were areas with medium potential. Second, in 2010–2015, compared with the PIMCI in Guizhou Province in the early 1990s, except for in Bijie, Anshun, and Qianxinan, there were no significant changes in potential, and the potential of the remaining cities decreased and some became areas with no potential.

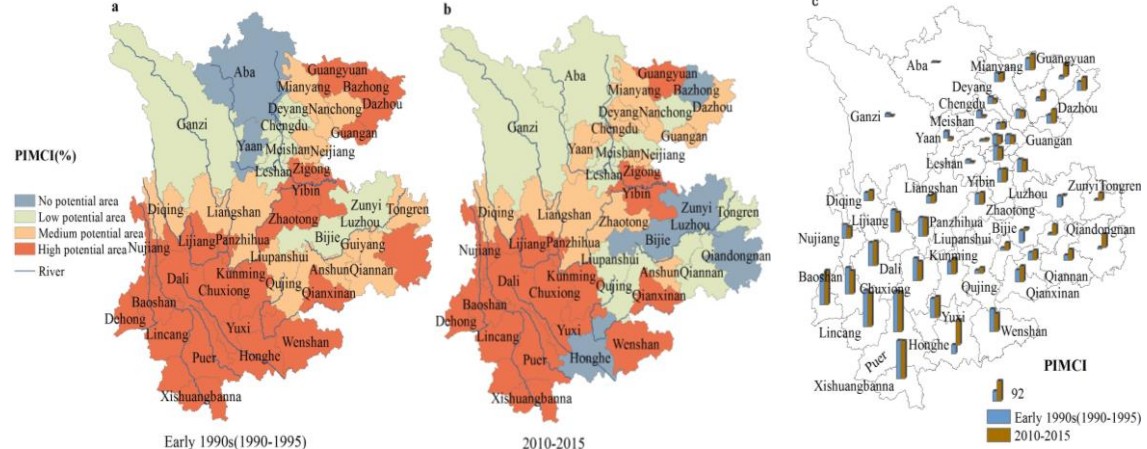

**Figure 4.** The characteristics of the PIMCI in southwest China. Note: No potential area can increase the potential value (≤ 0%). A low potential area can increase the potential value (0%~25%). A medium potential area can increase the potential value (25%~50%). A high potential area can increase the potential value (≥50%).

## 4. Discussion

### 4.1. Impact of Climate Change on the MCI

The results of the correlation between the MCI and climate factors are shown in Table 1. The linear relationship between the daily average temperature and the MCI in the southwestern region exhibited a significant increase ($p < 0.01$). Furthermore, the annual precipitation, precipitation intensity, and climatic aridity index had slight effects on the MCI. From 2006 to 2010, the planting structure in Honghe of Yunnan was adjusted to make full use of the light and heat conditions in autumn and winter; thus, agriculture was developed in winter. In the low-altitude, hot valley area, ratooning rice, intercropping, and multiple cropping were vigorously promoted in the northern planting dam area to achieve three crops a year [40], which was the reason for the highest change in the MCI in Honge in Yunnan.

**Table 1.** Pearson correlation coefficient between the MCI and impact factors.

| Impact Factors | Coefficient |
| :---: | :---: |
| Precipitation | 0.06 |
| Temperature | 0.600 ** |
| Precipitation intensity | 0.49 |
| Aridity index | −0.24 |
| Relief amplitude | −0.59 ** |
| Population | 0.08 |
| Gross domestic production | −0.66 ** |
| Gross farm production | −0.63 ** |
| Grain production | −0.33 |

Note: ** indicates significant at $p < 0.01$ level.

The changes in the spatial patterns of the daily average precipitation and temperature in the early 1990s to the early 2000s and 2010–2015 in the eastern and southern regions were significantly higher than those in the central and northwestern regions. However, the spatial distribution of the MCI was generally higher in the east and lower in the west, which was consistent with the trend of the temperature changes. Figure 5d–f indicates that the climate continued to warm over the last 25 years, leading to a spatial pattern of the PMCI in which the values in the eastern and southern regions were significantly higher than those in the central and northwestern regions. Overall, the spatial differences in the PMCI were obvious, and the temporal changes in the PMCI were not significant. The distribution pattern of the PMCI was mainly affected by the climate resources. Diqing is located in the southeastern margin of the Qinghai-Tibet Plateau, which is in the hinterland of the Hengduan Mountains and has a temperate and cold temperate climate with abundant water resources; however, the climate was cold, and rain was frequent in the rainy season [41]. Due to the abundant precipitation and insufficient heat, the PMCI of precipitation was high, and the PMCI of cumulative temperature was low, which ultimately caused the lowest PMCI value in the region. This region is the only one in which crops are grown only once a year in Yunnan Province, which was consistent with the results of Tao Wenxing's research on the characteristics of the PMCI [42]. The main reasons for the low PMCI were that the area was at a high altitude and high latitude, the water and heat conditions were poor, and the potential for re-cultivation of cultivated land was limited.

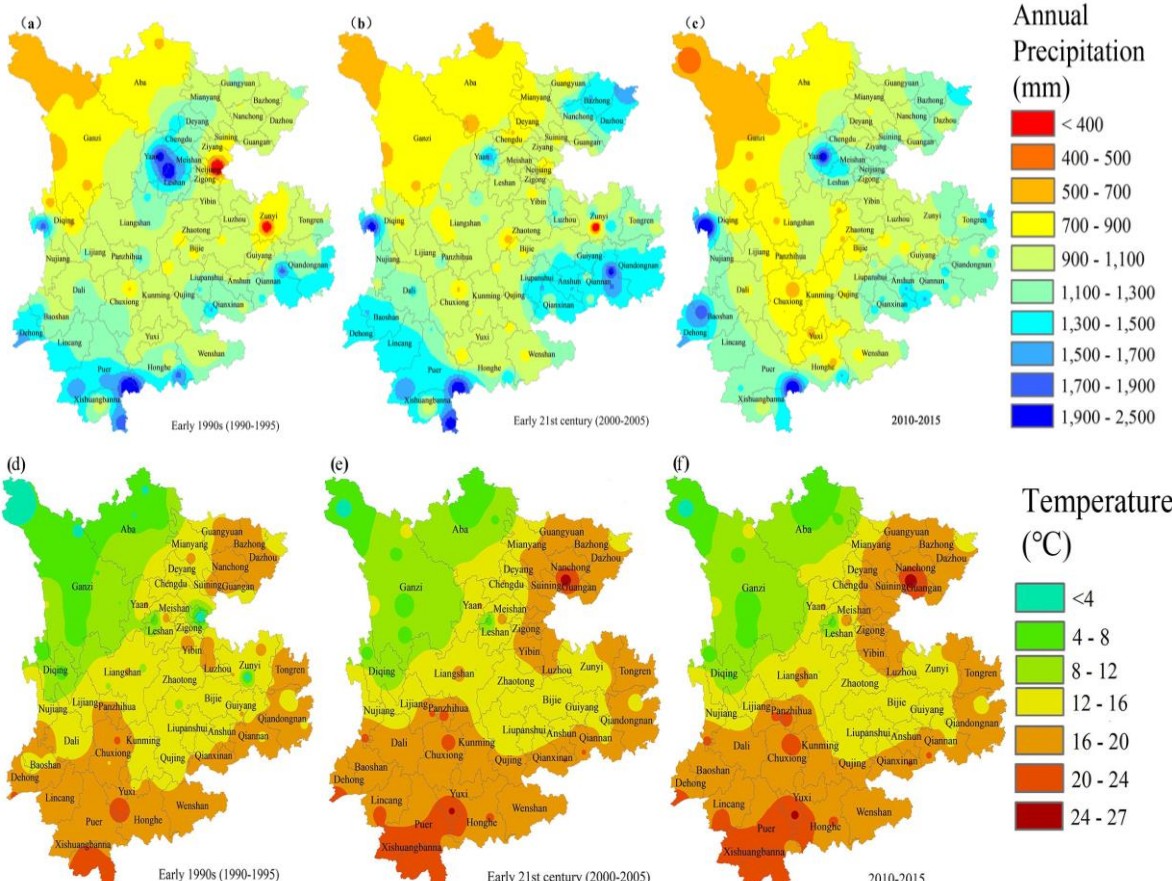

**Figure 5.** Spatial distribution of annual average precipitation and temperature changes from 1990 to 2015.

The southwestern area of China has frequent natural disasters. Figure 5 shows that the spatial and temporal distributions of water and heat resources in the southwestern region were uneven from the early 1990s to 2010 to 2015. In particular, the distribution of precipitation was extremely uneven in different regions (Figure 5a–c). The change in precipitation was the direct cause of regional drought.

The overall distribution characteristics of precipitation and temperature in southwest China were high in the south, low in the north, high in the east, and low in the west. Compared with the early 1990s, in the early 21st century, the temperature increased and precipitation decreased significantly in Sichuan and Yunnan provinces during 2010–2015, which led to a decrease in soil moisture in some areas, further increasing the frequency and range of drought. From the early 21st century to 2010–2015, the MCI declined in most areas of Sichuan Province and Yunnan Province, and the reductions in Chuxiong and Lincang in Yunnan Province were the most obvious. From 2006 to 2015, the MCI of the grain crops in the three provinces fluctuated with the following inter-annual variation rates: Sichuan > Guizhou > Yunnan, with values of −41.01%, −33.55%, and −29.60%, respectively. The reason for the decline in the MCI during this period was mainly due to the seriousness of various natural disasters in the southwestern region in 2006 and between the fall of 2009 and the spring of 2010; as a result, the no-harvest areas reached $83.4 \times 10^4$ hm$^2$ and $155 \times 10^4$ hm$^2$, respectively. Furthermore, the drought-affected areas reached $340.9 \times 10^4$ hm$^2$ and $350 \times 10^4$ hm$^2$, and the drought-induced areas reached $548.6 \times 10^4$ hm$^2$ and $503 \times 10^4$ hm$^2$ (Table 2). The no-harvest rate reached the maximum value in nearly 60 years at this time. Due to repeated disasters, some farmland, electricity, and fundamental water facilities were severely damaged, and this damage dampened the enthusiasm of farmers to some extent. Therefore, there was a decrease in the MCI during this period [43]. As previously mentioned, the MCI of cultivated land, the grain crop MCI and the cash crop MCI in Yunnan Province decreased sharply in 2010, so Yunnan was characterized as a center of severe drought in 2010, and drought disasters were serious. The area affected by drought exceeded $400 \times 10^4$ hm$^2$, which was the main reason for the reduction in the MCI of different types of crops [44]. Similarly, due to the increase in natural disasters, the grain crop MCI of Guizhou declined slightly in 2006 [45]. Because cash crops have high economic value, farmers were more enthusiastic about planting these crops. In recent years, the cash crop MCI has maintained a steady growth trend. In addition, there was a rainstorm disaster from 1994 to 1995 that spanned 331,200 hm$^2$ in Yunnan Province, which resulted in a reduction in the crop yield of 463,900 t [46], and the cash crop MCI decreased by 5.6%. Overall, the cash crop MCI in Yunnan showed double-peak stable growth; in contrast, the cash crop MCI in Guizhou Province showed fluctuating growth; however, the cash crop MCI inn Sichuan showed single-peak growth.

**Table 2.** Severe drought years in southwest China in 1949–2012 ($10^4$ hm$^2$) (data from literature) [47].

| Year | Drought-Induced Areas | Drought-Affected Areas | No-Harvest Areas | Grain Yield Per Hectare (kg) | Disaster Centre |
|------|------|------|------|------|------|
| 1979 | 427.9 | 210.0 | 48.1 | 166.6 | Guizhou |
| 1992 | 431.6 | 254.2 | 41.7 | 223.9 | Guizhou |
| 2001 | 566.0 | 280.1 | 89.0 | 221.4 | Sichuan |
| 2006 | 548.6 | 340.9 | 83.4 | 227.0 | Sichuan |
| 2010 | 503.3 | 350.4 | 155.3 | 257.4 | Yunnan |

*4.2. Impact of Human Activities on the MCI*

4.2.1. Impact of Land Use and Cover Changes

Land use and cover changes (LUCC) are one of the main types of human activities that can directly or indirectly influence the MCI via positive or negative effects. The spatial distribution of land use types has a clear regional pattern in southwest China, which was deeply influenced by climate, geomorphology, and hydrothermal distribution. The dominant land use types in southwest China were forest (46%), grass (28%), and farmland (23%) during 1980–2015 (Table 3). Figure 6 shows the spatial distribution of land use types from 1990 to 2015. Forest and grass were distributed primarily in the southern and eastern mountain areas, especially the grasslands in high mountain areas (e.g., Sichuan Plateau), while farmland was prevalent in basin areas. The areas of farmland and forestland decreased by 5371 km$^2$ and 2329 km$^2$, respectively. Grassland increased by 2178 km$^2$, and the urban

area increased by 582 km². The transition matrix for the land use type during 1990–2015 is illustrated in Table 4. The area of forest that was converted to grass was 70,849 km², which was greater than the transition from grass to forest (67,388 km²). There was a total of 26,047 km² of farmland area that was converted to grass, which was slightly less than the grass area that was converted to farmland (26,661 km²). The transition between grass and other ecosystems was obviously higher than the transition from other land use types, especially in comparison with forest. The urban area in 205 increased by 12% in comparison with the area in 1990; at the same time, the area of farmland in 2015 decreased by 2.3% in comparison with that in 1990. Rapid industrialization and urbanization are the main factors that reduce farmland areas, which finally had a negative effect on the MCI [48,49]. Both increasing urban area and decreasing farmland area had significant negative impacts on the MCI, especially in Yunnan Province, which showed a fluctuating downward trend and sudden declines in Guizhou Province in 2004–2006 and 2012–2014. However, in Sichuan Province, the MCI exhibited a slow increasing trend, but, the increases were relatively low. Our findings agree with the investigation of, who indicated that increasing the area of farmland was more effective than reclamation. Furthermore, the creation of desert areas will result in environmental problems such as soil erosion [45,50]. Moreover, rapid urbanization led to increasing labor costs, which is indeed the key driving force for MCI changes [51]. The temporal and spatial changes in the farmland pattern play a vital role in influencing the dynamic changes in the MCI. Thus, it is urgently needed to study to increase in crop production to ensure food security, which is significantly impacted by the contribution of the MCI to the Chinese grain yield potential [52].

**Table 3.** The land use changes in southwest China during the period 1990 and 2015 (Unit: Km²).

|  | Farmland | Forestland | Grassland | Water and Wetland | Urban | Others |
|---|---|---|---|---|---|---|
| 1990 | 240,836 | 480,486 | 289,704 | 10,461 | 4837 | 16,238 |
| 1995 | 234,031 | 486,652 | 289,802 | 10,198 | 5248 | 16,484 |
| 2000 | 237,471 | 478,690 | 292,297 | 10,867 | 6647 | 16,057 |
| 2005 | 238,293 | 478,179 | 292,570 | 10,721 | 6287 | 15,980 |
| 2010 | 237,471 | 478,691 | 292,299 | 10,867 | 6647 | 16,057 |
| 2015 | 235,431 | 477,852 | 291,707 | 11,703 | 9279 | 16,060 |
| Percentage % | 22.8 | 46.0 | 28.0 | 1.0 | 0.6 | 1.5 |

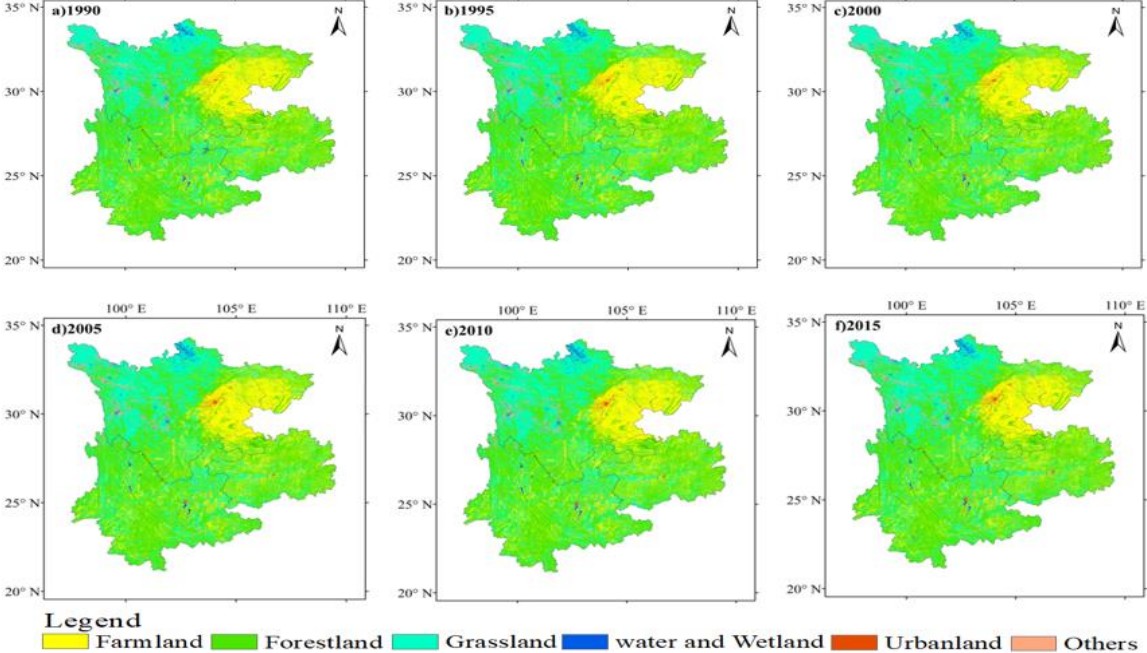

**Figure 6.** Spatial distribution of land use changes from 1990 to 2015.

**Table 4.** Transformation matrix for the land use type in southwest China during 1990 and 2015 (Unit: Km$^2$).

| 1990 | 2015 | | | | | | |
|---|---|---|---|---|---|---|---|
| | Farmland | Forestland | Grassland | Water & Wetland | Urban | Others | Total 1990 |
| Farmland | 143,825 | 62,843 | 26,047 | 2530 | 5418 | 95 | 240,758 |
| Forestland | 60,380 | 344,967 | 70,849 | 1283 | 1195 | 1398 | 480,072 |
| Grassland | 26,661 | 67,388 | 185,698 | 3285 | 955 | 5489 | 289,476 |
| Water & wetland | 2070 | 1024 | 2686 | 4212 | 270 | 179 | 10,441 |
| Urban | 2379 | 470 | 371 | 171 | 1433 | 12 | 4836 |
| Others | 72 | 1051 | 6003 | 218 | 8 | 8884 | 16,236 |
| Total 2015 | 235,387 | 477,743 | 291,654 | 11,699 | 5418 | 16,057 | |

### 4.2.2. Impact of Socioeconomic Development

Due to the rapid increase in socioeconomic developments, which include population, gross domestic production, gross farm production, and grain production, in southwest China, the MCI in this region has been greatly impacted. These variables are important indices for changing the land use land cover pattern and abundance of water resources and finally influencing the MCI, so these indices play a major role in controlling the MCI, especially in fast-developing countries such as China. Figure 7 shows the temporal changes in the population, gross domestic production, gross farm production, and grain production during the period from 1990–2015. Population exhibited an increasing trend in Yunnan, Sichuan, and Guizhou, and the increase was especially evident in Yunnan, which had a significant increasing trend of $40.82 \times 10^6$ person/year (Table 5). The average total population of southwest China increased with an annual rate of 0.29% per year during the period from 1990–2015. On the one hand, gross domestic production exhibited a significant increasing trend in Sichuan of $959 \times 10^8$ Yuan/year; however, it had a nonsignificant increasing trend in Yunnan and Guizhou provinces, and the average total gross domestic production increased with an annual rate of 9.8% per year during the study period. Gross farm production exhibited an obvious increasing trend in Yunnan and Sichuan provinces with increases of 48.3 and $98 \times 10^9$ Yuan, respectively. The increasing trend shown clearly after 2006 was sharp; in contrast, grain production exhibited an increasing trend in Yunnan and Guizhou provinces, especially in Yunnan, which had a significant increasing trend of $35.4 \times 10^6$ tons/year. The average total gross farm production and grain production of Yunnan and Sichuan provinces increased with annual rates of 7.4% per year and 0.8% per year, respectively, during 1990–2015.

**Table 5.** Mann–Kendall rank correlation trend test of population, gross farm production, and grain production in Yunnan, Sichuan, and Guizhou.

| Indices | Province | MK Test | | |
|---|---|---|---|---|
| | | Z | B | Trend |
| Population | Yunnan | 7.14 | 40.82 | + |
| | Sichuan | 0.62 | 2.18 | No trend |
| | Guizhou | 1.19 | 6.96 | No trend |
| | Average | 4.36 | 15.54 | + |
| Gross farm production | Yunnan | 6.97 | 48.25 | + |
| | Sichuan | 6.61 | 97.78 | + |
| | Guizhou | 5.16 | 12.42 | + |
| | Average | 6.44 | 54.01 | + |
| Grain production | Yunnan | 6.48 | 35.40 | + |
| | Sichuan | 0.13 | 0.75 | No trend |
| | Guizhou | 3.66 | 11.00 | + |
| | Average | 4.23 | 15.30 | + |

Note: + means significant increasing change trend (*P* > 95%), No trend: means no significant change trend (*P* < 80%).

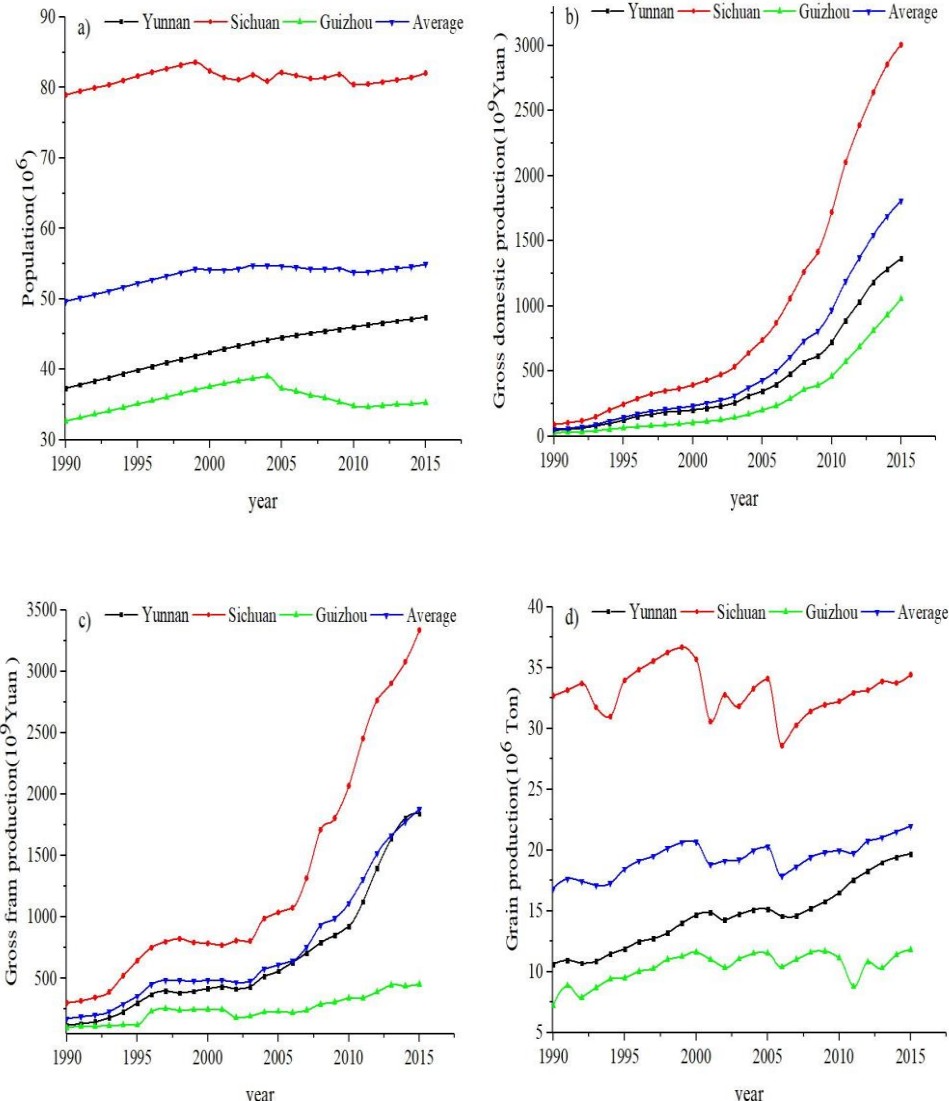

**Figure 7.** Temporal changes of socioeconomics variables in southwest China.

Based on the above results of socioeconomic development, the increased gross farm production and grain production are good examples of the increasing MCI in southwest China, especially in Sichuan Province, which is covered by more farmland than Yunnan and Guizhou provinces. The correlation coefficient between the MCI and gross domestic production indicated a significant negative correlation due to the increasing gross domestic production resulting from increasing economic growth and rapid urbanization, which ultimately had a negative impact on the farmland area and MCI. Population had a positive effect on the MCI due to increases in the population's demand for agricultural products, which encouraged farmers to increase crop acreage, the MCI, and agricultural production to obtain greater economic benefits. Our results agree with the previous results of [53], who documented that an increasing MCI played an important role in grain production. Furthermore, the grain yield level and agricultural production conditions were the main factors affecting the dynamic changes in the MCI [52]. The MCI is the most effective way to increase grain production; thus, increasing production in southwest China is evidence of an increasing trend in the MCI [54]. With population growth, agricultural science and technology progress, and the implementation of various agricultural policies, the development and utilization of farmland in these areas has increased, and the implementation of multiple cropping has increased. Due to the increasing population and economic growth resulting in

increasing demands for grain products and further decreases in the farmland area in China, the MCI has become a key process for increasing land production.

### 4.3. The Influence of Terrain Factors on the MCI

The distribution of relief amplitude and soil type in southwestern China is shown in Figure 8. It can be seen from the figure that the eastern part of Sichuan was mainly plains, and the soil type was mostly purplish soils, which were concentrated and distributed in fragments. The soil had high nutrients and excellent production performance. Crops were mainly distributed in the plains of Sichuan. Therefore, the largest MCI of Sichuan Province was mainly distributed in the eastern part of Sichuan. However, the terrain of the plateau in the northwest was highly fluctuating, and the soil types were mostly frigid frozen soils and felty soils, resulting in a relatively low MCI in the west and north, and there were no significant changes over the years. In addition to the small relief amplitude, purplish soils and high MCI in the central region of Yunnan Province, most of the regions in Yunnan and Guizhou provinces were dominated by platform and hills, and most of the soil types were red soil and yellow soils. Crops were mainly distributed in mountain basins and in thousands of valley plains. Based on this result, the change in the MCI at different relief amplitudes is clear (Figure 9). The MCI fluctuated greatly in the three provinces of southwest China. From Guizhou to Sichuan to Yunnan, the MCI showed a slight fluctuation and decreasing trend. Both the MCI and the relief amplitude fluctuated less in Guizhou Province than in Sichuan Province. The average relief amplitude of each city in Yunnan Province was higher than that of each city in the other two provinces. The figure shows that the MCI and relief amplitude had opposite relationships, such as those in Ganzi and Aba of Sichuan Province and Lincang and Puer of Yunnan Province, and the regions with low MCI values had large relief amplitudes. On the other hand, the areas with higher MCIs had small relief amplitudes, such as Chengdu and Nanchong of Sichuan Province and Qiandongnan and Zunyi of Guizhou Province. This phenomenon was particularly obvious in Sichuan Province. The overall reduction in the MCI in Sichuan Province was mainly distributed in Chengdu and its surrounding areas. According to the research, from 2000 to 2015, the area of sloping farmland in Sichuan Province decreased by 3263 km$^2$. The areas of smooth slope farmland ($<10°$) decreased sharply by 1467 km$^2$, especially in Chengdu and its surrounding areas. The areas of steep slope farmland ($>25°$) decreased by 302 km$^2$, still accounting for the largest proportion of cultivated land area throughout the region. Terrain changes were the main cause of the reductions in the MCI in this region [55]. Furthermore, correlation analysis showed that there was a significant decreasing exponential relationship between the relief amplitude and MCI in the three provinces ($y = 326.77e^{-0.01x}$, $R^2 = 0.3336$, $P < 0.01$), which further proved that the MCI decreased as the relief amplitude increased.

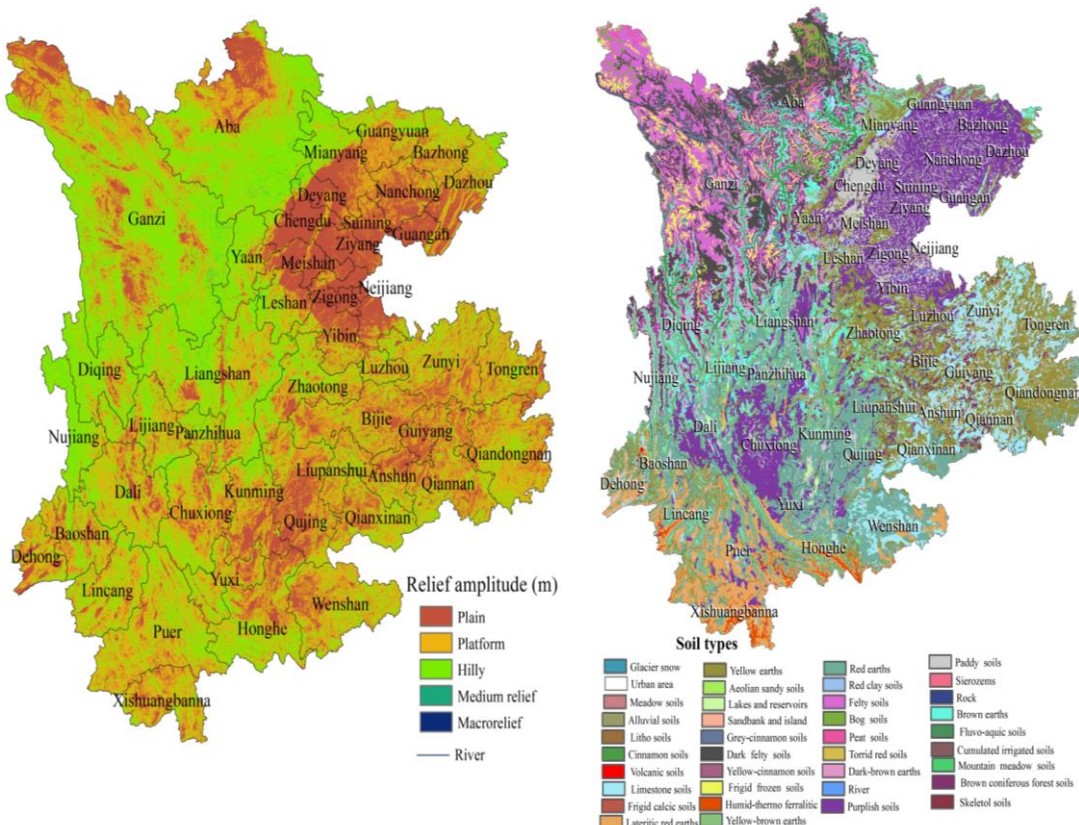

**Figure 8.** Distribution characteristics of relief amplitude and soil type in southwest China.

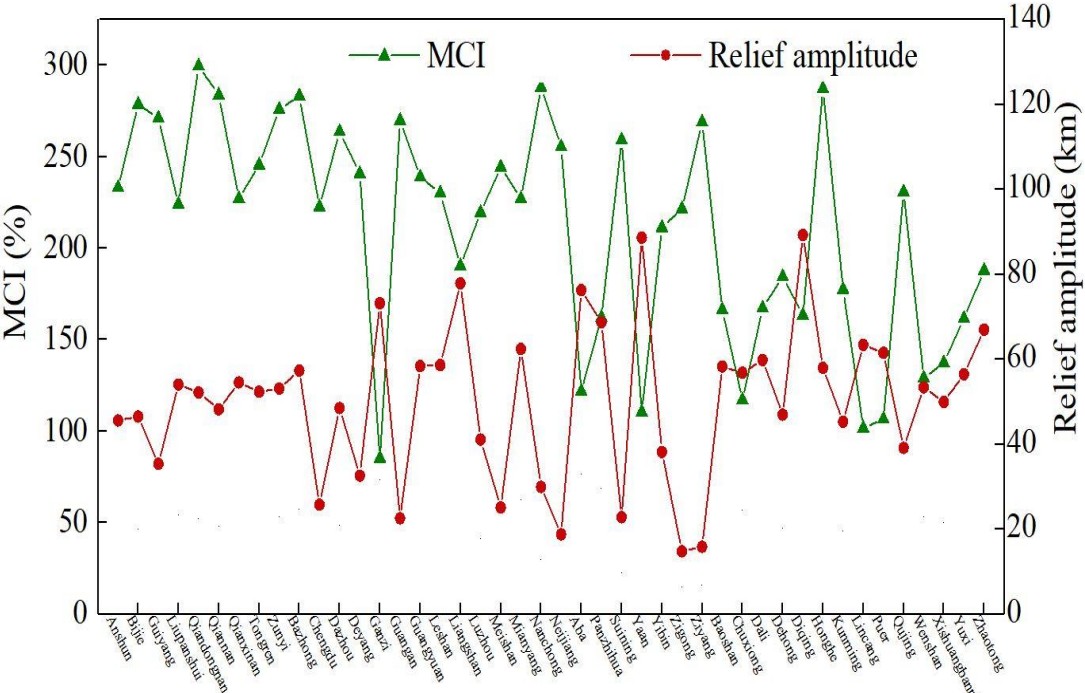

**Figure 9.** Variation characteristics of the MCI in different relief amplitudes.

## 5. Conclusions and Policy Suggestions

*5.1. Conclusions*

The study's goal was to estimate the PMCIs of districts and evaluate the regional distribution of the PIMCI in different areas by analyzing the overall annual characteristics and spatial variations in the MCI and analyzing the factors influencing the MCI, so the main conclusions are as follows:

(1) Temporal characteristics of the variations in the MCI: In Yunnan Province, the MCI showed a fluctuating downward trend with sudden declines in Guizhou Province from 2004–2006 and 2012–2014, while in Sichuan Province, it exhibited an increasing trend. The grain crop MCI was roughly divided into two stages, from 1990 to 2006, it decreased slowly in Yunnan but the downward trend was sharp in 2000, while both Guizhou and Sichuan provinces showed increasing trends, with the growth rate in Sichuan being relatively slow. The greatest change in the three provinces occurred from 2006 to 2015. The inter-annual variations in the grain crop MCIs were as follows: Sichuan > Guizhou > Yunnan, with values of −41.01%, −33.55%, and −29.60%, respectively. In contrast, the cash crop MCI has maintained a steady growth trend in recent years.

(2) Spatial characteristics of the variations in the MCI: The internal distribution of the MCI in the southwestern region as a whole was high in the east and low in the west throughout the study period. Moreover, the MCI was higher in the central region than in the northern and southern regions. Particularly, from the early 1990s to the early 21st century, the MCI increased in most areas of the three provinces, and the increase was particularly obvious in Guizhou Province, while in the early 21st century, the MCI of each city and state in the three provinces reached the maximum value. In 2010–2015, the MCI declined in most areas of Sichuan Province, especially in Chengdu and its surrounding areas. In the southern part of Yunnan Province, the MCI was the lowest in Xishuangbanna and Pu'er city, and the phenomenon of abandoned farmland was more serious in these areas. The overall level of arable cropping in Yunnan Province was lower than that in Guizhou Province and Sichuan Province.

(3) For the PMCI, the spatial differences in the PMCI in southwest China were obvious, and the temporal differences were not significant. In the early 1990s and 2010–2015, the PMCI was significantly higher in the eastern and southern regions than in the central and northwestern regions. The PMCI was largest in the eastern part of Sichuan, which was much higher than that in the western region. The PMCI was relatively large in all regions of Yunnan due to the large space for the full utilization of light, heat, and water resources. Over the past 25 years, as climate warming continued to increase, the average PMCI for all Sichuan showed an increasing trend, except in Bazhong in the eastern part of Sichuan, which exhibited a decreasing trend. However, the PMCI in the eastern part of Yunnan showed a declining trend, especially in Honghe, and other regions maintained high potential. In contrast, the changes in the other cities in Guizhou Province were not significant over the past 25 years, except for the obvious changes in Qiannan and Zunyi. The PIMCI values in Yunnan Province were significantly higher than those in Guizhou and Sichuan provinces, especially in the southern part of Yunnan, which was the smallest area with high potential in Guizhou Province. However, Yunnan Province was rich in water and heat resources, but the overall utilization of cultivated land was inefficient.

(4) Factors influencing the MCI: Climate change and natural disasters had a certain influence on the MCI changes in southwest China, especially temperature changes, which was the key factor driving the MCI changes. The temporal and spatial changes in land use, such as farmland and urbanization, played a vital role in influencing the dynamic changes in the MCI. The spatial variations in the MCI were closely related to the terrain, as the MCI in the hilly area decreased, and the MCI in the plain area increased.

Finally, in our paper, human factors may have interfered with the statistical methods. Therefore, in further studies, we will combine remote sensing data and statistical data to consider other factors, such as agricultural facilities, other social economies, and other variables, to improve the quantitative accuracy of the MCI and provide a more scientific theoretical basis for the effective utilization of cultivated land resources and the optimal allocation of agriculture in various regions.

*5.2. Policy Suggestions*

(1) The trend of the MCI in southwest China decreased, especially after 2006, due to environmental problems such as floods, serious drought, and extreme temperatures, which impacted the infrastructures of farmland, water conservation, transportation, and urbanization; thus, the government and policy makers should devote more concern to the development of production conditions and water-saving technology to conserve water resources in the rainy seasons and relieve drought in the dry seasons. Furthermore, the government must support rain-flood utilization, seawater desalination, and effective drought management practices to face future water resource shrinkages.

(2) The variations in the MCI in southwest China are mostly due to the differences between the three provinces, which including differences in the water resource, heat, fertilizer, and farmland characteristics. The MCI exhibited a slight decreasing trend, especially in Yunnan Province, due to the labor shortage problems and cost of farm production. Therefore, the government should pay attention to the promotion of science and technology to solve the labor shortage problems by accelerating the construction of labor-saving technology, implementing large-scale operations, agricultural mechanization, and farmland management. China's science and technology promotion rate is currently only approximately 35%, with large room for development.

(3) Both non-agricultural industry and per capita farmland have strong positive effects on the MCI, so policy makers should transfer the excess rural labor to the secondary and tertiary industries, alleviate pressure on farmlands, and raise the gross farm production, which will ultimately support farmers in less developed regions and increase the grain production and MCI in southwest China [53].

**Author Contributions:** Conceptualization, C.Z. and H.H.; methodology, C.Z.; software, C.Z.; validation, C.Z.; formal analysis, A.M.; investigation, C.Z. and A.M.; resources, C.Z.; data curation, C.Z.; writing original draft preparation, C.Z.; writing review and editing C.Z.; visualization, C.Z.; supervision, H.H.; project administration, H.H.; funding acquisition, H.H.

**Funding:** This work was financially supported by the National Key Research and Development Program, Grant/Award number (2017YFC0505205), and National Natural Science Foundation of China (41672180).

**Conflicts of Interest:** The authors declare no conflict of interest.

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
