# Peer review of "The Impact of Climate Change and Human Activity on Spatiotemporal Patterns of Multiple Cropping Index in South West China"

_sustainability, doi:10.3390/su11195308_

Round 1

Reviewer 1 Report

The research is very interesting and economically friendly,  economically important. The spatio-temporal pattern and trends of MCI changes in 1990–2015 in southwestern China were analyzed using a quantitative model. The spatio-temporal pattern and trends of MCI changes in 1990–2015 in southwestern China were analyzed using a quantitative model. In addition, analyzed relationships between maximum MCI and heat and water resources using theoretical PMCI formulas and space-time PIMCI formulas. In addition, the relationship between maximum MCI and heat and water resources was analyzed using theoretical PMCI formulas and space-time PIMCI formulas. Previews of attempts to calculate PMCI districts and assess the regional growth range of many different pruning (PIMCI) in various China operations by analyzing the entire characteristics and diversity of MCI spatial configurations and calculating the multiple framing index factor. An attempt was made to estimate PMCI districts and assess the regional distribution of growth potential of many indicators pruning (PIMCI) in various areas of China by examining the entire characteristics and diversity of MCI spatial patterns and analyzing the impact of multiple framing index factors. The spatio-temporal pattern and trends of MCI changes in 1990–2015 in southwestern China were analyzed using a quantitative model. In addition, the relationship between maximum MCI and heat and water resources was analyzed using theoretical PMCI formulas and space-time PIMCI formulas. An attempt was made to estimate PMCI districts and assess the regional distribution of growth potential of multiple pruning indicators (PIMCI) in various areas of China by examining the totality of features and diversity of MCI spatial patterns and analyzing the impact of multiple framing index factors. Work assumptions, theses are correct. The research methodology is extensively and comprehensively described. Test results well documented and collected. A discussion of the results could be better treated. Spatio-temporal pattern and trends of changes in MCI in 1990–2015 in southwestern China were analyzed by applying a quantitative model. In addition, the relationship between maximum MCI and heat and water resources was analyzed using theoretical PMCI and space-time PIMCI formulas. An attempt was made to estimate the districts PMCI and assess the regional distribution of the growth potential of multiple pruning indices (PIMCI) in various areas of China by examining the whole of the characteristics and diversity of spatial patterns of MCI and analyzing the impact of factors of multiple framing index.
Work assumptions, theses are correct. The research methodology is extensively and comprehensively described. Test results well documented and described. The results discussion could have been better treated. Honest conclusions. Literature items selected, selected and used. Improve English at work and make better use of literature.

Author Response

Point 1: Does the introduction provide sufficient background and include all relevant references? 

Response 1: Thanks for your invaluable suggestions. We have rewritten the introduction section in order to improve it, please see the “introduction section” for details.

Point 2: Is the research design appropriate?

Response 2: Thanks for your suggestions. We have reorganized the paper to be in an appropriate design.

Point 3: Honest conclusions. Literature items selected, selected and used. Improve English at work and make better use of literature.

Response 3: Thanks for your suggestions. We have added more English literature to improve the literature review, furthermore, we have improved English in the whole of the paper by English editing company and ourselves, please check the “text file(Sustainability_582673, Final version 201909012)” and we have added the certificate language in the “cover letter” for details.

We have improved English in the whole of the paper by English editing company and ourselves

Reviewer 2 Report

The main aim of the paper is to analyze the spatial and temporal variations of the multiple cropping index and study the influence factors of multiple cropping index.

The document presented a sound theoretical concept (MCI), that can add a new dimension of measure for productivity assessment/comparison in various agro-ecologies.

The paper shows interesting information, but the explanations about PMCI are referred to papers in chinese that are difficult to find and to understand. A recommendation for a further explanation is given, so the paper can be properly assessed. Without theses explanations is difficult to assess the paper.

Specific comments

Line 35. Reference needed.

Author Response

Point 1: The explanations about PMCI are referred to papers in Chinese that are difficult to find and to understand

Response 1: Thanks for your suggestions. We have reexplained the PMCI by international literature to as well as Chinese literature to be easy to find and to understand, please see “introduction section” in the second paragraph for details.

Point 2: Line 35. Reference needed.

Response 2: Thanks for your comments. We have dereferenced the whole paper, please check the text for details.

Reviewer 3 Report

Some suggestions for minor improvement are indicated in sticky notes in the pdf file of the manuscript.

Author Response

Point 1: Some suggestions for minor improvement are indicated in sticky notes in the pdf file of the manuscript.

Response 1: Thanks for your invaluable comments. We have replied for all your comments in text file, please check it for details,examples are as follows

This paragraph is a summary, not necessary here.

To comprehensively evaluate the intensive utilization of cultivated land resources in southwestern China, we have analyzed the spatiotemporal pattern and change trend of the MCI from 1990 to 2015 in Southwest China through using the model of the quantitative relationship between the maximum MCI and heat and water resources, furthermore, analyzed the theoretical PMCI and spatiotemporal patterns of the PIMCI to achieve our goal which is to estimate the PMCI of districts and evaluated the regional distribution of the potential increment of multiple cropping index (PIMCI) in different areas by studying the overall annual characteristics and spatial pattern variations of the MCI and analyze the influence factors of multiple cropping index, so the main conclusions were follows as:

After the modification is

The study’s goal was to estimate the PMCIs of districts and evaluate the regional distribution of the PIMCI in different areas by analyzing the overall annual characteristics and spatial variation

Round 2

Reviewer 2 Report

The main aim of the paper is to analyze the spatial and temporal variations of the multiple cropping index and study the influence factors of multiple cropping index.

The paper shows interesting information, several improvements have done but the still the main concerns are maintained. The papers that are referred to define MCI and PMCI use a different terminology. Why has been changed? And the main concern is the calculation of PMCI. Formulas are given in line 210 and the following lines, but why those values? Where they come from? The papers that should explain that data are in chines so it is difficult to understand the explanation. In that case a detailed explanation of this data should be given in the text. In that way the reader can assess the data that is given.

Author Response

Response to Reviewers Comments Sustainability (MS: sustainability-582673)

Dear Editors and Reviewers:

Thank you for your letter and for the reviewers comments concerning our manuscript entitled “The Impact of Climate Change and Human Activities on Spatiotemporal Patterns of Multiple Cropping Index in South West China” (ID: sustainability-582673). Those comments are all valuable and very helpful for revising and improving our paper, as well as the important guiding significance to our researches. We have studied comments carefully and have made correction which we hope meet with approval. Revised portion are marked in revisions mode in the paper. The main corrections in the paper and the responds to the reviewer’s comments are as following:                                                                                                                      

Responds to the reviewer’s comments:

The paper shows interesting information, several improvements have done but the still the main concerns are maintained. The papers that are referred to define MCI and PMCI use a different terminology. Why has been changed? And the main concern is the calculation of PMCI. Formulas are given in line 210 and the following lines, but why those values? Where they come from? The papers that should explain that data are in chines so it is difficult to understand the explanation. In that case a detailed explanation of this data should be given in the text. In that way the reader can assess the data that is given.

Re: Point 1:Definition of MCI and PMCI

(1) MCI According to Zuo et al. (2014), Multiple Cropping Index(MCI), the number of crops planted per year in the same piece of land, repre-senting the actual situation of the multiple cropping systems. MCI is an approach used to increase the planting frequency of crops based on the original crop planting; it makes full use of idle fields and improves land use efficiency.The MCI is calculated on the administrative division unit using the statistics of crop planting area and cultivated land area.which is an observation of the real cropping situation (Dalrymple,1971). We revised as suggested by the reviewer (see line 57)

(2) PMCI According to Zuo et al. (2014), potential multiple cropping index (PMCI), the maximum number of crops planted per year in the same piece of land, representing the potential capacity for the implementation of multiple crop-ping. During the growth and development of any crops, certain water and heat conditions are required. In reality, the maximum MCI results from making full use of the available light, heat and water resources. According to Gao et al. (2014), the PMCI of cumulative temperature and the PMCI of precipitation were calculated using Equations (2) and (3), respectively. Finally, the minimum values of both the PMCI of precipitation and the PMCI of cumulative temperature were collected for each pixel, and the PMCI was calculated using Equation (4) We revised as suggested by the reviewer (see line 91,92,93,94,95and Method)

Re:Point 2:PMCI parameter value range

According to Gao et al. (2014).The relational model between MCI and water& heat. Based on the regional heat and precipitation conditions, the model calculated the potential multiple cropping index under the constraints of the two factors, taking the minimum of the two as the regional potential multiple cropping index. When calculating the the PMCI of cumulative temperature, the four obvious boundaries of the change of multiple cropping index were determined according to the temperature difference ≥accumulated temperature 3400℃, 4200℃, 5200℃ and 6200℃; when calculating the PMCI of precipitation, used 500mm and 1 200mm as the dividing line of potential prediction of multiple cropping index. The study used the average annual precipitation and the average activity accumulated temperature as model inputs to calculate the average potential multiple cropping index. The calculation model is as follows

(2)

                          (3)

 PMCI=MIN(MT,MR)                                                                         (4)

PMCI is the potential multiple cropping index, MT is the PMCI of cumulative temperature (%), and MR is the PMCI of precipitation (%). T is the accumulated temperature greater than 0 (°C). R is the average annual precipitation (mm).

Re:Point 3:The difference between MCI and PMCI

The MCI is the observed multiple cropping value that occurs in real cultivated land, and it is usually used to measure the degree of intensive utilization of cultivated land resources and evaluate the basic situation of the utilization of cultivated land resources. We revised as suggested by the reviewer (see line 76,77,78)

The potential multiple cropping index (PMCI) is the maximum MCI that can be achieved by making full use of natural resources such as water, soil, light and heat in multiple cropping patterns. PMCI is the maximum number of crops planted per year in the same piece of land, representing the potential capacity for the implementation of multiple crop-ping, it is the theoretical value for predicting the potential for multiple cropping ability of cultivated land. We revised as suggested by the reviewer (see line 91,92,93,94,95)

The actual multiple cropping index refers to the actual re-cultivation level that can be achieved under the constraints of economic, policy, manpower, and technical conditions. Affected by the natural ecological environment and socio-economic conditions, the multiple cropping index and actual multiple cropping index of a region will constantly change.

Re:Point 4:Why choose these two indexes separately?

The multiple cropping index is a basic indicator for measuring the intensive use of cultivated land resources in the study of farming systems. It reflects the extent to which agricultural production uses agricultural production resources on a time scale.The value of the multiple cropping index is large, indicating that the number of multiple croppings is high, and the degree of utilization of cultivated land is high. On the contrary, the value of the multiple cropping index is small, indicating that the number of multiple cropping is less and the degree of utilization of cultivated land is low.During the growth and development of any crops, certain water and heat conditions are required.100-300 days of growth period,1000-3000℃of accumulated temperature and 300-800mm of precipitation are the basic requirements for their growth and maturity. Otherwise, the crops cannot complete the whole life. Only when the growth period, accumulated temperature and precipitation of a region exceed the needs of its leading crops, other crops can be planted according to residual growth period, accumulated temperature and precipitation.Therefore, the potential multiple cropping index (PMCI) is the result of the region's full use of light, heat and water resources. After decades of rapid development of agricultural production in China, the level of productivity has been greatly improved, which has led to the reform of the cropping system. At present, the planting system in many places of southwestern China has been implemented under the condition of high multiple cropping index.By comparing the theoretical predictions (PIMCI) with the actual observations (MCI), it is possible to more clearly formulate more practical countermeasures for the adjustment of planting structure in the later stage of agriculture in this area.

We tried our best to improve the manuscript and made some changes in the manuscript. These changes will not influence the content and framework of the paper. And here we did not list the changes but marked in red in revised paper.

We appreciate for Editors/Reviewers’ warm work earnestly, and hope that the correction will meet with approval.

       Finally, we have professionally polished the full-text language. as shown in the following figure.

Once again, thank you very much for your comments and suggestions.
